# An Entropy Analysis-Based Window Size Optimization Scheme for Merging LiDAR Data Frames

**DOI:** 10.3390/s22239293

**Published:** 2022-11-29

**Authors:** Taesik Kim, Jinman Jung, Hong Min, Young-Hoon Jung

**Affiliations:** 1Department of Civil Engineering, Hongik University, Seoul 04066, Republic of Korea; 2Department of Computer Engineering, Inha University, Incheon 22212, Republic of Korea; 3School of Computing, Gachon University, Seongnam 13120, Republic of Korea; 4Department of Civil Engineering, Kyung Hee University, Yongin 17104, Republic of Korea

**Keywords:** LiDAR, entropy analysis, window size optimization, merging point cloud data frames, linear structure extraction

## Abstract

LiDAR is a useful technology for gathering point cloud data from its environment and has been adapted to many applications. We use a cost-efficient LiDAR system attached to a moving object to estimate the location of the moving object using referenced linear structures. In the stationary state, the accuracy of extracting linear structures is low given the low-cost LiDAR. We propose a merging scheme for the LiDAR data frames to improve the accuracy by using the movement of the moving object. The proposed scheme tries to find the optimal window size by means of an entropy analysis. The optimal window size is determined by finding the minimum point between the entropy indicator of the ideal result and the entropy indicator of the actual result of each window size. The proposed indicator can describe the accuracy of the entire path of the moving object at each window size using a simple single value. The experimental results show that the proposed scheme can improve the linear structure extraction accuracy.

## 1. Introduction

Light Detection and Ranging (LiDAR) technologies have been rapidly developed and there are various applications based on LiDAR. LiDAR systems are classified into three types: spatial, spectral, and temporal information capturing [1]. Spatial schemes obtain point cloud data based on Time of Flight (ToF) measurements and spectral schemes measure the information of a material using what is termed Laser Return Intensity (LRI). Temporal schemes gather additional information based on spatial and spectral information using a repeated LiDAR technique. Application developers must select an optimal product because each LiDAR type has different functionalities and specifications such as range, Field of View (FoV), precision, and accuracy.

In mobile environments, LiDAR has extended application areas and most autonomous vehicles are equipped with LiDAR. LiDAR technology is essential for Advanced Driver-Assistance Systems (ADAS) that handle automatically steering, accelerating, and braking under the driver’s supervision [2]. Autonomous driving vehicles use LiDAR sensors, which provide high-resolution and real-time 3D representation data to detect the surrounding environment and obstacles. Odometry is essential for accurate self-localization for path planning and environment perception, which are key features related to driving safety [3]. LiDAR-based odometry can handle environmental variations by taking advantage of its active sensor emitting laser beams. The main functionality of LiDAR odometry is a registration between the current scan data and the reference point cloud data, which is solved by Iterative Closed Point (IPC) algorithm. 

Simultaneous Localization and Mapping (SLAM) is also a promising field related to mobile LiDAR. It is designed to build or update a map of an unknown environment while simultaneously keeping track of an agent’s location. LiDAR is a more popular mechanism in SLAM compared to other mechanisms such as radar and ultra-wideband positioning due to LiDAR’s high precision, wide coverage, and longevity [4]. Traditional LiDAR-based SLAM algorithms mainly leverage the geometric features from the scene context, while the intensity information from LiDAR is ignored. The SLAM framework, which uses both geometry and intensity information for odometry estimations, provides reliable and accurate localization in multiple environments and outperforms geometric-only methods [5]. LiDAR-based SLAM can also be used in indoor navigation systems of autonomous vehicles because LiDAR-based SLAM can provide more robust localization than image-based SLAM in a lack textured environment [6].

In this paper, we use a mounted LiDAR system on a moving object to monitor an excavation site in an urban area. We extract linear structures that have references to the location information to calibrate the accuracy of a satellite-based navigation system. The location of a moving object is measured through the triangulation method using the extracted linear structures. The Velodyne Puck equipped with our system has relatively low specifications, as shown in Table 1. As the distance between the Velodyne Puck and linear structures increases, the extraction accuracy rapidly decreases due to its low vertical angular resolution. Therefore, we propose a sliding window mechanism to improve the accuracy of collected data from the low-spec LiDAR system. The proposed sliding window mechanism merges consecutive frames to acquire more point cloud data and improves the possibility of linear structure extraction by considering the movement of the mobile object.

We also propose an optimal window size decision algorithm based on Shannon entropy analysis. We calculate the entropy of the desired extracted linear structures at each point as a reference structure to find the optimal window size. Upon a change in the location of the moving object, our algorithm recalculates the entropy of each point with a change in the window size. The optimal window size of each point is set to the minimum difference between the reference entropy and the recalculated entropy. Our experimental results show that the proposed sliding window mechanism and the optimal window size decision algorithm perform well.

The rest of this paper is organized as follows. Section 2 describes related works. Section 3 presents the proposed entropy analysis-based window size optimization scheme. Section 4 presents the evaluation results, and Section 5 concludes the paper.

## 2. Related Works

### 2.1. Mobile LiDAR Systems

There are many systems that can monitor or map environments using LiDAR in a mobile device. A road segmentation-based curb detection method was proposed to provide navigation information for a self-driving vehicle with LiDAR [7]. A sliding-beam method is used to distinguish on-road and off-road areas and split segments of a road from point cloud data. A curb-detection method has also been applied to determine the positions of curbs for different road segments.

Zhang et al. proposed a real-time localization method that estimates the location of an autonomous driving vehicle using a 3D-LiDAR system [8]. Point cloud data of a curb beside the road are extracted based on the laser intensity features and matched to a high-precision curb map generated offline. A map-matching method between the point cloud data collected from the 3D-LiDAR system and the high-precision curb map is designed using an Iterative Closest Point (ICP) algorithm to improve the accuracy of the vehicle’s location. The authors verified the performance of the proposed system by comparing the estimated location with that of a low-cost global positioning system.

LiDAR can also be used to detect and trace various traffic participants, such as vehicles, pedestrians, obstacles, and bicycles, to guarantee the safety of self-driving vehicles [9]. This type of tracking system consists of three modules: mask generation, depth estimation, and a retracking mechanism. The retracking mechanism overcomes repetitive appearances and disappearances of objects caused by the movement of a self-driving vehicle and traffic participants. LiDAR-based object detection systems perform well under harsh weather conditions such as heavy rain and dense fog [10].

Cheng et al. proposed a water leakage detector for use in shielded tunnels that relies on deep learning with mobile LiDAR intensity images [11]. The mobile LiDAR system is designed to collect point cloud data and intensity information simultaneously from the shielded tunnels. The intensity images based on the collected point cloud data are generated and used for training with a Fully Convolutional Network (FCN) to improve the accuracy of the water leakage detection process. After training with the FCN, water leaks in the shielded tunnels can be extracted through intensity image semantic segmentation. 

Luo et al. proposed an intelligent detection method for the spraying of tunnel shotcrete [12]. In this method, LiDAR can obtain a 3D model of the tunnel and extract the positions of arches because spraying areas are usually divided by arches in a tunnel. A YOLO-based model is used to detect the approximate bounding boxes of the arches and a line-detection algorithm is used for determining the final spraying areas.

A road-marking extraction and classification scheme was also proposed [13]. The proposed framework applies capsule-based deep learning from a massive and unordered mobile laser scanner. First, this framework segments the road surface from 3D point cloud data and generates 2D georeferenced intensity images. Then, a U-shaped capsule-based network model is used to extract road markings based on convolutional and deconvolutional capsule operations. Finally, a hybrid capsule-based network model is applied to classify different types of road markings using a revised dynamic routing algorithm and a large-margin Softmax loss function.

Silva et al. proposed a robust fusion type of LiDAR system for mobile robots [14]. Mobile robots must fuse heterogeneous data because these robots are equipped with various positioning sensors such as LiDAR, radar, ultrasound sensors, and cameras. A geometrical model in the form of a Gaussian Process regression-based resolution matching algorithm is used to align the LiDAR and camera data spatially. 

LiDAR can also be utilized in an unmanned aerial vehicle (UAV) to monitor numerous wide areas simultaneously. Hu et al. proposed an UAV-LiDAR system that monitors forest ecosystems and manages forest resources [15]. The authors analyzed the performance of the UAV equipped with various LiDAR sensors, such as the RIEGL VUX-1, HESAI Pandar40, Velodyne Puck Lite, among others. The UAV flies at different altitude and speed combinations. The authors attempted to discover the usefulness of their low-cost UAV-LiDAR system, despite its weaknesses, including low-intensity data and a narrow FoV.

Many SLAM systems use 3D LiDAR to collect data quickly. Park et al. presented a map-centric SLAM solution with improved accuracy and effectiveness outcomes given its use of fusion-based mapping and deformation-based loop closure schemes [16]. The proposed system uses a local continuous time, surface resolution preserving matching algorithm, a normal-inverse-Wishart-based surface element fusion model, and a robust metric loop closure model to achieve accuracy and effectiveness. 

Karimi et al. proposed a low-latency LiDAR SLAM using continuous scan slicing and concurrent matching to support real-time indoor navigation [17]. The continuous scan slicing splits point cloud data from a rotating LiDAR in a concurrent multi-threaded matching pipeline for 6D pose estimation with a high update rate and low latency. 

A plane adjustment approach was also proposed in the SLAM field with LiDAR [18]. Plane adjustment combines optimizing plane parameters and LiDAR poses to achieve improved accuracy outcomes. The proposed system consists of three components: localization, local mapping, and global mapping. Localization establishes the association between local and global data. Local and global mapping improve the quality of the map via plane adjustments.

### 2.2. Point Cloud Data Merging

Chen et al. proposed a range merging scheme [19]. The proposed scheme can reconstruct a high-density point cloud using a type of point cloud error optimization based on depth computations and confidence estimations. The depth map computation is equivalent to minimizing a type of energy equation. A confidence estimation is used to eliminate outliers for each depth map.

Morita et al. presented a map generation and merging method that uses a mobile laser scanner based on the Normal Distributions Transform (NDT) scan matching with a full graph-based SLAM [20]. The NDT scan matching based recursive SLAM generates a point cloud map until the loop is detected. The generated submaps of different small areas are merged such that the Euclidian distance between two consecutive submaps is minimized.

Merging ground and aerial point cloud data was also proposed [21]. These two types of data are separately obtained from ground and aerial LiDAR. This approach attempts to find point matches between two images from different sources with different viewpoints and scales. Two scenes are matched using a sparse mesh and calibrated by a geometrical consistency check. Finally, the point clouds are merged via bundle adjustment by linking the ground to aerial tracks. 

Point cloud registration refers to the process of finding a spatial transformation between two point cloud datasets. Serafin et al. proposed an extension to the well-known ICP called the Normal Iterative Closest Point (NICP) [22]. NICP uses a sophisticated error metric that considers the distances between corresponding points and the corresponding surface normal point. 

A dynamic segment merging scheme was proposed to identify the non-photosynthetic components of trees by semantically segmenting tree point clouds and examining the linear shape prior to each segment [23]. The authors define a similarity metric, which is estimated for each segment, using this metric to merge similar neighboring segments in a step-by-step manner. Non-photosynthetic segments such as stems and branches are identified by estimating the linear feature of the trees. 

There is also a hybrid approach that merges 3D point cloud data from LiDAR and generated aerial photographs [24]. This approach aligns the coordinates and scales to merge point cloud data generated by the UAV with point cloud generated by the LiDAR system. The merged point cloud data show reasonable accuracy, and the accuracy can be improved by using data acquisition optimization and post-processing steps.

The goal of these merging schemes is to build a global map by connecting adjacent submaps or to calibrate point cloud data from different data sources. However, our approach aims to improve the quality of point cloud data from a low-power LiDAR system using merging consecutive point cloud frames. We also determine the optimal number of frames by considering the movement of a mobile object.

## 3. Entropy Analysis based Window Size Optimization Scheme

### 3.1. Application Scenario

Figure 1 shows an overview of the proposed mobile position system based on linear structure extraction from point cloud data collected using 3D-LiDAR. A mobile object equipped with LiDAR collects point cloud data while moving through the monitoring space. If there are obstacles such as tall buildings or street trees near the monitoring space, the accuracy of the satellite-based positioning system is reduced. The proposed system extracts a linear structure that serves as a reference from the collected point cloud data to correct the position of the mobile object. When three or more linear structures are extracted, the distance between the linear structure and the mobile object can be measured. The position of the mobile object can be estimated through trilateration.

Figure 2 shows the hardware components of our system. There are three main components: a Jetson Xavier as the processing module, a Velodyne Puck as the LiDAR module, and a LPMS-USBAL2 unit as the IMU (Inertial Measurement Unit). The Jetson Xavier has an 8-core ARM v8.2 64-bit architecture-based CPU, a 512-core NVIDIA volta architecture-based GPU, 16 GB of memory, 32 GB of storage space, and an additional 1 TB of removable storage space to save data. The aforementioned Velodyne puck is connected to the Jetson Xavier via a 100 Mbps LAN connection. The LPMS-USBAL2 unit connected to the Jetson Xavier with USB has roll and yaw of ±180°, a pitch of ±90°, and a resolution of 0.01°. The accuracy of the IMU is 0.5° and 2° in static and dynamic environments, respectively. 

Figure 3 shows the workflow of the proposed positioning system. First, point cloud data are recorded in the Robot Operating System (ROS) BAG file format at regular intervals in LiDAR. The recorded BAG file is then converted to a PCD (Point Cloud Data) file in which one frame is saved, and calibration is conducted by referring to the IMU data. Linear structures are extracted from the calibrated PCD file, and if the number of extracted linear structures is three or less, the PCD file is merged with the next consecutive frame file to increase the point cloud data, after which the linear structures are extracted again. When four or more linear structures are extracted, the distance is calculated through a comparison with the reference linear structures, and the position of the mobile object is measured based on the calculated result.

### 3.2. System Modeling

In our model, we assume that we know the location information of all reference linear structures and that the mobile object repeatedly traverses the monitored area along the driving route without stopping. Points of interest requiring accurate location information can be randomly located and are mainly targeted at areas where satellite-based location signals are not captured or at areas where large errors occur, even when signals are captured. Figure 4 shows the process of extracting a linear structure at each point of interest from a mobile object equipped with a LiDAR system with a limited detection range. When the mobile object arrives at a point of interest while moving in the monitored area along the detection path, the linear structure is extracted and the ID of the extracted linear structure is acquired.

Figure 5a shows the ID list of the ideal linear structure extracted from the point of interest along the moving path of the mobile object. All of the linear structures that should be detected are extracted from the results, and there are no false detection results. However, if a linear structure is extracted using the actual point cloud data measured while the mobile object is moving, it may contain results that were not extracted or that are false positives (F/P), as shown in Figure 5b. Therefore, an indicator is needed to determine how accurate the actual result is compared to the ideal result.

We applied the Shannon entropy analysis-based window size optimization scheme proposed by Wu et al. [25] to our system to evaluate the actual accuracy of the reference linear structure extraction results at the points of interest. 

Table 2 describes the notations used in this paper.

According to the Shannon entropy definition, the indicator of system *E(X)* is defined as Equation (1).
(1)$$E\left(X\right)=-\overset{n}{\underset{i=0}{\sum }}P\left(x_{i}\right)logP\left(x_{i}\right),\;where\;n\;is\;the\;number\;of\;points$$

We defined $P\left(x_{i}\right)$ as the probability of a correctly detected linear structure at point *i*, as derived from the relationship between the total number of extracted linear structures of all points of interest in the ideal result and the number of correctly detected linear structures in the actual result at point *i*. This relationship is defined as Equation (2).
(2)$$P\left(x_{i}\right)=\frac{\left|C_{i}-F_{i}\right|}{N}\;,\;where\;N=\overset{n}{\underset{i=1}{\sum }}C_{i}\;in\;ideal\;result$$

Our goal function described in Equation (3) is to find the minimum point between the entropy indicator of the ideal result and the entropy indicator of the actual result with a window size of *W*.
(3)$$\min\left(\left|E_{R}\left(X\right)-E_{W}\left(X\right)\right|\right),\;where\;2\;\leq \;W\leq 10)\;$$

### 3.3. Window Size Optimization Algorithm

Algorithm 1 shows the pseudocode for the proposed window size optimization algorithm based on the entropy analysis. These input parameters include the set of point of interests (*I*), the total number of point of interests (*T_I_*), the set of referenced linear structures’ IDs (*R*), and the total number of extracted linear structures at each point of interest in the ideal result (*N*). First, the entropy indicator of the ideal result (*E_R_(X)*) is calculated using N and C*_i_* and initializes the minimum value of the difference between *E_R_(X)* and the entropy indicator of the actual result at window size *i* (*E_i_(X)*). Next, the entropy indicator at each point of interest is repeatedly accumulated by *T_I_*. When the absolute value of the current difference between *E_R_(X)* and *E_i_(X)* is less than *V_min_*, *V_min_* and the optimal window size (*ω*) are updated. Finally, when the loop ends, we can find the optimal window size, *ω*, and the algorithm returns the *ω* value.
**Algorithm 1** Finding Optimal Window SizeInput: *I, T_I_, R, N*Output: Optimal window size *ω*1Calculate *E_R_(X)*2*V_min_* = INF3**for** *i* = 2 **to** 10 **do**4     *E_i_(X)* = 05     **for** *j* = 1 **to** *T_I_* **do**6     
*E_i_(X) +=*
$P\left(x_{j}\right)*\log\frac{1}{P\left(x_{j}\right)}\;\;,\;\;\;where\;x_{j}\in \;I$
7     
**end for**
8     **if**$\left|E_{R}\left(X\right)-E_{i}\left(X\right)\right|$ < *V_min_*
**then**9     
*V_min_*
$=\left|E_{R}\left(X\right)-E_{i}\left(X\right)\right|$
10     *ω* = *i*11**end if**12**end for**13**return** *ω*

## 4. Evaluation Results

### 4.1. Experimental Environment

Figure 6 depicts the appearance of a moving vehicle equipped with a LiDAR device (Velodyne Puck), an edge computing platform, and an IMU. We run the test vehicle on a test field to estimate the location of the vehicle using the proposed scheme.

Figure 7 shows the preparation process including the location measurements of the referenced linear structures and points of interest using a high-accuracy GNSS (Global Navigation Satellite System) device, in this case, a Trimble R10. The Trimble R10 is a high-accuracy position measurement device that has a horizontal error of less than 2 cm and a vertical error of 5 cm during stationary measurements.

### 4.2. Effects of the Proposed Window Mechanism

When we extract linear structures from a single frame (without the proposed window mechanism), the accuracy of linear structure extraction is low due to the low density of the Velodyne Puck. When applying the proposed window mechanism, the accuracy of linear structure extraction increases. As shown in Figure 8, when we set the window size to 3, the contours of trees, traffic signs, and streetlamps are clear due to the merging consecutive point cloud data. The merged point cloud data, which reflect the vertical movements of the moving vehicle, can improve the extraction accuracy of the linear structure extraction process.

However, the proposed window mechanism has a side effect, as shown in Figure 9. As the number of merged point cloud data frames increases, the error also increases. This accumulated error makes it difficult to extract linear structures. Therefore, it is important to measure the effect of the proposed window mechanism and to determine the optimal window size.

### 4.3. Entropy Indicator Comparison

We analyzed the entropy indicators of a single frame with a static window size (*W* = 3) and with the optimal window size for the same route for vehicles moving at about 2.5 km/h (Normal) and about 5 km/h (Fast). Along the moving path, there are 12 reference linear structures and five points of interest. Figure 10 shows the entropy indicators of three schemes at different speeds of the vehicle.

For the single frame case (a), the entropy indicator is high at each point of interest because faulty linear structures are detected and referenced linear structures are missed due to the low density of the point cloud data. At the static window size (b), the entropy indicator fluctuates regardless of whether the static window size is identical to the optimal window size. When the static window size closes to the optimal window size, the entropy indicator approaches zero. When the static window size is further from the optimal window size, the entropy indicator is higher. The optimal window size (c) shows the lowest entropy indicator value at every point of interest. In the comparison of the three schemes in terms of the total entropy indicator values (d), the optimal window size scheme shows the lowest value. The speed does not have a significant effect on the entropy indicator values of the three schemes.

Figure 11 shows the entropy indicator values of three schemes with the movements of the vehicle. We control the moving vehicle by having it follow a set path (Normal) and by having it drive in a zigzag direction (Zigzag). When the vehicle drives severely from side to side, the entropy indicator values of all schemes increase dramatically because the point cloud data are distorted. Accumulating frames with heavy noise data makes it difficult to extract linear structures.

Figure 12 shows the proposed scheme overhead in terms of the PCD file size, merging time, and the linear structure execution time. In the case of merging time, it is stable as increasing the number of frames. However, merged PCD file size linearly increases and the linear structure exponentially increases execution time. The weak point of the proposed scheme is the high overhead during the merging process.

## 5. Conclusions

We used a low-power LiDAR system attached to a moving object to estimate the location of the moving object. The movement of the moving object creates differences in each collected point cloud data instance at every moment. We applied a data frame merging scheme to improve the accuracy of linear structures. The proposed window scheme calculates a single indicator to describe the effect of the window size on the entire path of the moving object using entropy analysis. We also show various experimental results to verify the accuracy improvement of the proposed scheme. Our future works will include the development of a dynamic optimization algorithm that determines the optimal result at each point of interest and a technique to mitigate scattered point cloud data when merging data frames during fast movements. We will also attempt to apply IMU-based calibration methods for the future system to reduce noise from raw point cloud data.

## Figures and Tables

**Figure 1 sensors-22-09293-f001:**
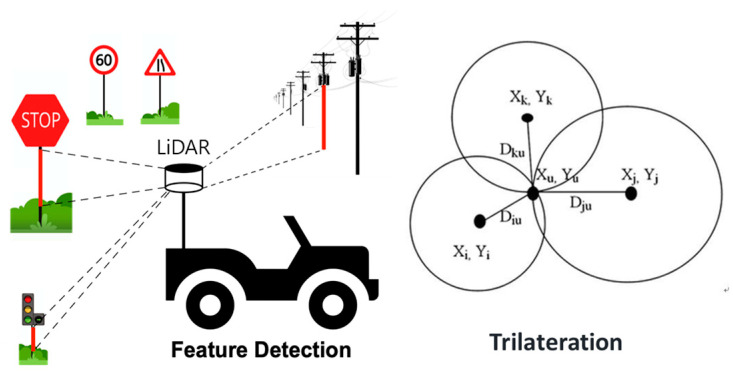
Mobile Positioning System based on Linear Structures Extraction.

**Figure 2 sensors-22-09293-f002:**
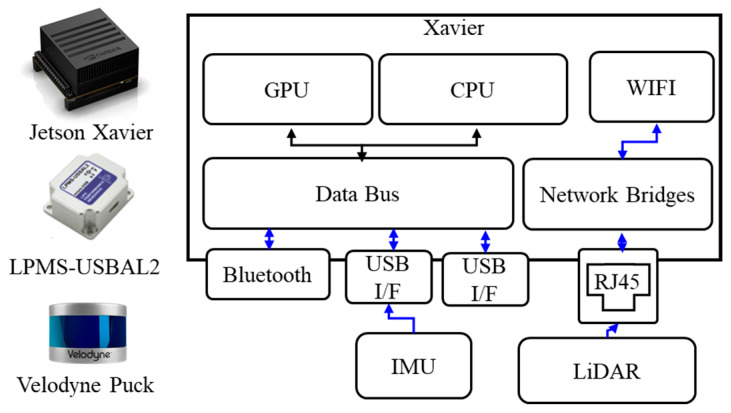
Hardware Components of the Proposed System.

**Figure 3 sensors-22-09293-f003:**
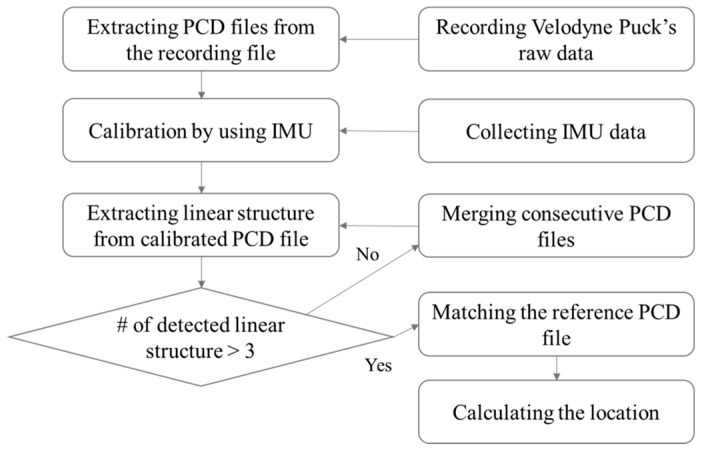
Workflow of the Proposed System.

**Figure 4 sensors-22-09293-f004:**
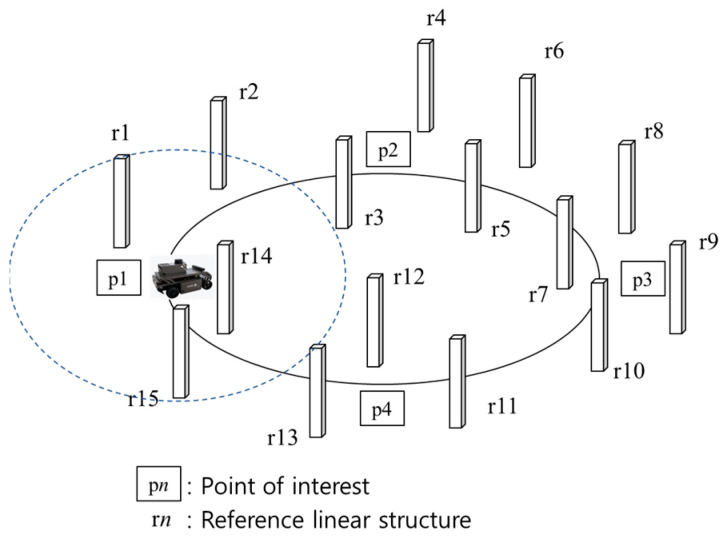
Linear Structure Extraction at Point of Interests.

**Figure 5 sensors-22-09293-f005:**
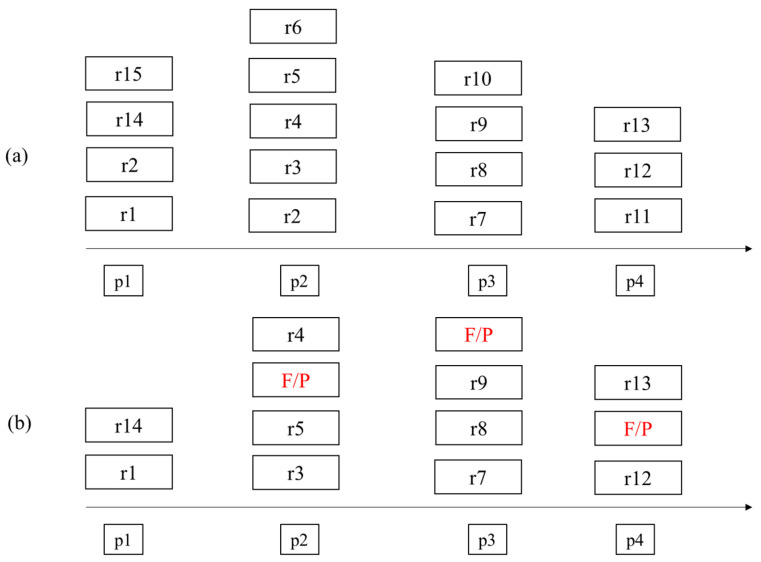
Extracted References at Point of Interest. (**a**) Ideal result (**b**) Actual result (F/P is False Positive).

**Figure 6 sensors-22-09293-f006:**
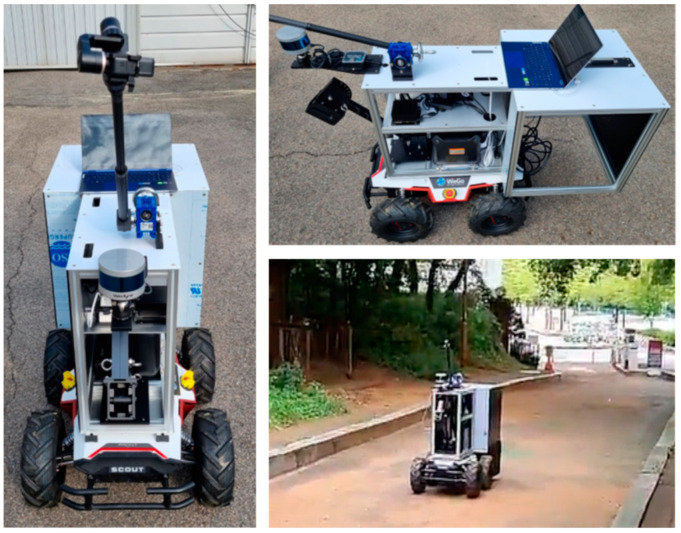
Test Vehicle Appearance and Test Field Run.

**Figure 7 sensors-22-09293-f007:**
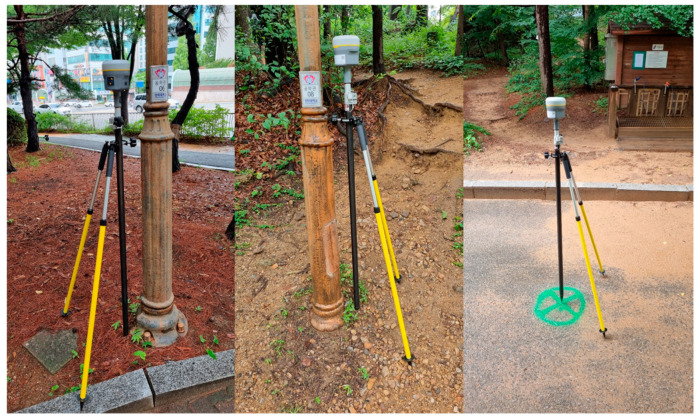
Trimble R10 Measurements.

**Figure 8 sensors-22-09293-f008:**
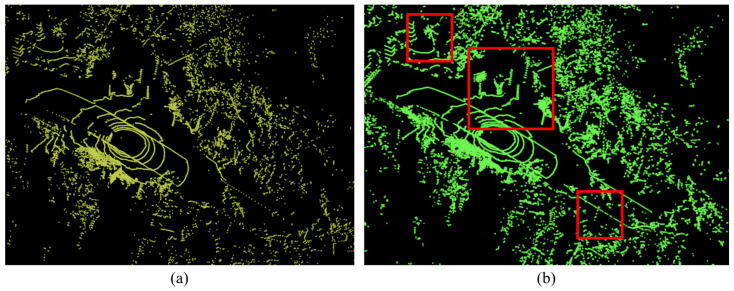
Comparison between (**a**) Without and (**b**) With Window Mechanism. Red box areas noticeably improve the shape of linear structures.

**Figure 9 sensors-22-09293-f009:**
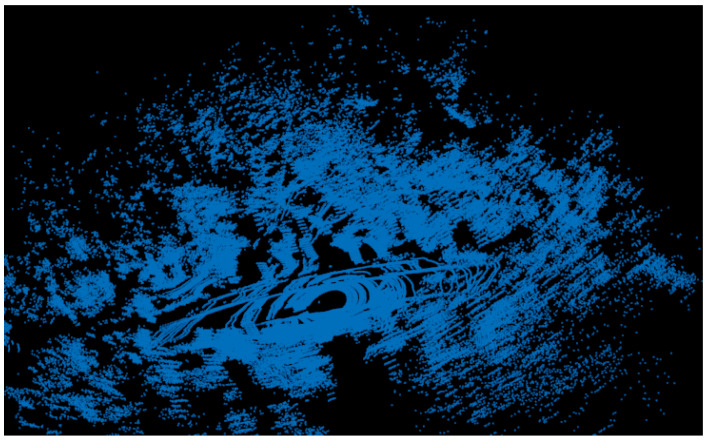
Side Effect of the Proposed Window Mechanism.

**Figure 10 sensors-22-09293-f010:**
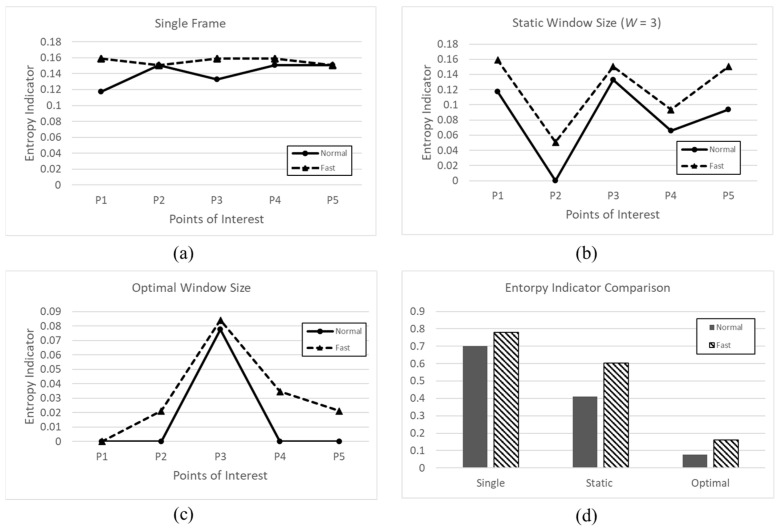
Entropy Indicator Values Comparison among the Three Schemes at Different Speeds. (**a**) Entropy indicator of single frame; (**b**) Entropy indicator of static window size; (**c**) Entropy Indicator of optimal window size; (**d**) Entropy indicator comparison of each scheme.

**Figure 11 sensors-22-09293-f011:**
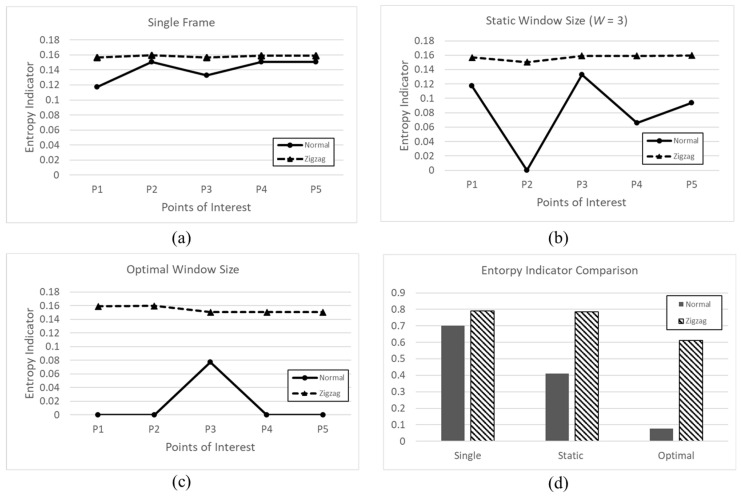
Entropy Indicator Values Comparison among the Three Schemes during Movement. (**a**) Entropy indicator of single frame; (**b**) Entropy indicator of static window size; (**c**) Entropy Indicator of optimal window size; (**d**) Entropy indicator comparison of each scheme.

**Figure 12 sensors-22-09293-f012:**
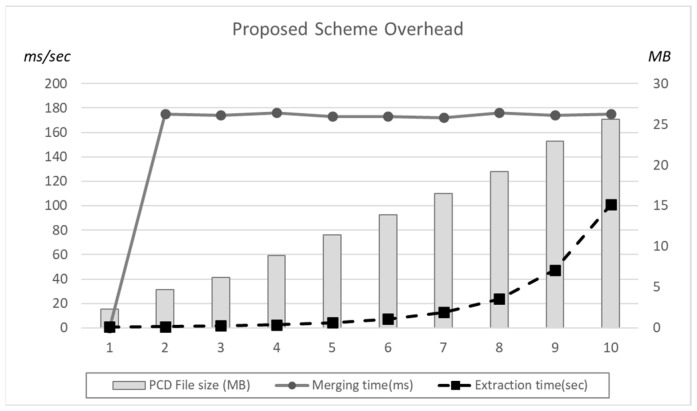
Proposed scheme Overhead in terms of File size and Execution Time.

**Table 1 sensors-22-09293-t001:** Velodyne Puck and Ultra Puck specifications.

	Velodyne Puck	Velodyne Ultra Puck
# of channels	16	32
Max range	100 m	200 m
Accuracy	±3 cm	±3 cm
FoV	30° (−15° to +15°)	40° (−25° to +15°)
Rotation rate	5~20 Hz	5~20 Hz
Vertical angular resolution	2°	0.33°
Horizontal angular resolution	0.1~0.4°	0.1~0.4°
# of frames	10	10
Weight	830 g	925 g

**Table 2 sensors-22-09293-t002:** Notations.

Notation	Description
*E(X)*	Indicator of system X’s entropy
*I*	Set of point of interests $\left(e.g.,\;I=\left\{p1,\;p2,\;p3,\;p4\right\}\right)$
*T_I_*	Total number of point of interests
*R*	Set of referenced linear structures’ ID $\left(e.g.,\;R=\left\{r1,\;r2,\;r3,\;\ldots r15\right\}\right)$
*W*	Window size (2 <= W <= 10)
N	Total number of extracted linear structure at each point of interest in ideal result (e.g., *N* = 16 in Figure 5a)
$C_{i}$	The number of detected linear structure at point *i* (*pi*)
$F_{i}$	The number of incorrectly detected linear structure at point *i* (*pi*)
$P\left(x_{i}\right)$	The probability of correctly detected linear structure at point *i* (*pi*)
*E_R_(X)*	Entropy indicator of ideal result
*E_W_(X)*	Entropy indicator of actual result at window size *W*

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
