# Peer review of "An Entropy Analysis-Based Window Size Optimization Scheme for Merging LiDAR Data Frames"

_sensors, 2022, doi:10.3390/s22239293_

Round 1

Reviewer 1 Report

The proposed window size optimization scheme is useful to improve the estimation accuracy of a vehicle's position especially where is not supporting GNSS mechanism. The proposed scheme is interesting and this paper is well-structured. The issue handled in this paper is suitable for the aim and scope of the special issue entitled Application of Semantic Technologies in Sensors and Sensing Systems. However, there are unclear explanations and incorrect expressions. This paper is carefully reviewed by a professional editor and needs to be reformatted by using sensor’s template. 

The proposed window size optimization scheme is useful to improve the estimation accuracy of a vehicle's position especially where is not supporting GNSS mechanism. The proposed scheme is interesting and this paper is well-structured. The issue handled in this paper is suitable for the aim and scope of the special issue for Application of Semantic Technologies in Sensors and Sensing Systems. 

  However, this paper has some issues and its contribution is not clear. Also, the paper has some unclear explanations and incorrect expressions.   - This paper should be carefully reviewed by a professional editor and needs to be reformatted according to the journal's template. - Where is novelty on this work? It’s not clearly explained. Lidar related studies focus on detecting objects by using collected point cloud data. In the case of MMS(Mobile Mapping Systems), many researchers have conducted studies to calibrate or erase the noise. The authors applied window-based mechanism to merging point cloud data frames and  entropy analysis to find optimal window size. Sliding window scheme is common in the network field and is applied to many applications.  - This paper needs more related works. - This paper needs to provide more explanations about the experiments. - There are some grammatical errors. They should be revised.

Author Response

Dear anonymous reviewer 1,

We thank you for your kind and careful review. The manuscript was revised according to your helpful comments as an attached file.

Please, find an attached reply file.

We highly appreciate your valuable comments again.

Reviewer 2 Report

This paper proposed an optimal window size decision algorithm to improve the accuracy of detecting referenced linear structure for finding the location of the moving vehicle. This paper suggested an interesting issue in positioning systems without supporting the GPS. Applying the window scheme to a LiDAR based localization system is the originality of this paper. This paper is well-structured but there are some comments as follows:

- Some sentences are not clear and difficult to understand. This paper must be reviewed by a native English speaker.

- In the experimental section, the author should provide results considering the speed and direction of the moving vehicle.

Author Response

Dear anonymous reviewer 2,

We thank you for your kind and careful review. The manuscript was revised according to your helpful comments as an attached file.

Please, find an attached reply file.

We highly appreciate your valuable comments again.

Reviewer 3 Report

The authors devised a new approach to determine the location information of a vehicle by using lidar. The authors explained the system design, the proposed window size optimization algorithm, and experimental results. The experimental vehicle is quite impressive and could be used in real-world applications. Investigating comprehensive related works can easily compare with previous works to the proposed study. The previous studies tried to reduce outliers from point cloud data and are difficult to use for moving vehicles. This study devises a window-based merging scheme and finds the optimal window size by considering the movement of a moving vehicle. The experimental result is also reasonable. There are some minor comments. Some expressions are not natural, so the authors must revise unnatural expressions. The authors only explained the advantages of the proposed scheme. I think that the proposed scheme has weaknesses in terms of processing overhead and memory usage. There are some IMU-based calibration algorithms to improve the accuracy of object extraction. The authors should consider applying an IMU-based calibration algorithm for the proposed scheme.

Author Response

Dear anonymous reviewer 3,

We thank you for your kind and careful review. The manuscript was revised according to your helpful comments as an attached file.

Please, find an attached reply file.

We highly appreciate your valuable comments again.
